# Two *Bacillus* spp. Strains Improve the Structure and Diversity of the Rhizosphere Soil Microbial Community of *Lilium brownii* var. *viridulum*

**DOI:** 10.3390/microorganisms11051229

**Published:** 2023-05-06

**Authors:** Jing Tu, Xin Zhao, Yuanru Yang, Yongjian Yi, Hongying Wang, Baoyang Wei, Liangbin Zeng

**Affiliations:** 1College of Bioscience and Biotechnology, Hunan Agricultural University, Changsha 410125, China; 2Institute of Bast Fiber Crops, Chinese Academy of Agricultural Sciences, Changsha 410205, China

**Keywords:** *Bacillus* spp., lily Fusarium wilt, rhizosphere soil, soil microbial diversity, high-throughput sequencing

## Abstract

Lily Fusarium wilt disease caused by *Fusarium* spp. spreads rapidly and is highly destructive, leading to a severe reduction in yield. In this study, lily (*Lilium brownii* var. *viridulum*) bulbs were irrigated after planting with suspensions of two *Bacillus* strains that effectively control lily *Fusarium* wilt disease to assess their effects on the rhizosphere soil properties and microbial community. A high-throughput sequencing of microorganisms in the rhizosphere soil was performed and the soil physicochemical properties were measured. The FunGuild and Tax4Fun tools were used for a functional profile prediction. The results showed that *Bacillus amyloliquefaciens* BF1 and *B. subtilis* Y37 controlled lily Fusarium wilt disease with control efficacies of 58.74% and 68.93%, respectively, and effectively colonized the rhizosphere soil. BF1 and Y37 increased the bacterial diversity and richness of the rhizosphere soil and improved the physicochemical properties of the soil, thereby favoring the proliferation of beneficial microbes. The relative abundance of beneficial bacteria was increased and that of pathogenic bacteria was decreased. *Bacillus* abundance in the rhizosphere was positively correlated with most soil physicochemical properties, whereas *Fusarium* abundance was negatively correlated with most physicochemical properties. Functional prediction revealed that irrigation with BF1 and Y37 significantly upregulated glycolysis/gluconeogenesis among metabolism and absorption pathways. This study provides insights into the mechanism by which two *Bacillus* strains with antifungal activity, BF1 and Y37, antagonize plant pathogenic fungi and lays the foundation for their effective application as biocontrol agents.

## 1. Introduction

Lily (*Lilium brownii*) is an economically important bulbous flower crop with high ornamental, edible, and medicinal value. Lily bulbs are rich in secondary metabolites, such as phenols, saponins, sterols, and alkaloids, which have sedative, anti-inflammatory, and antitussive activities [1,2]. Lily Fusarium wilt, a soil-borne fungal disease caused by *Fusarium* spp., has been a common disease affecting cultivated Longya (*Lilium brownii* var. *viridulum*) lily plants for many years. In addition, Fusarium wilt affects tomato, cucumber, and many other crop plants, and strongly impacts yields [3,4]. In addition to Fusarium wilt, lily plant growth and yield are also affected by diseases such as leaf spots and leaf scorch [5,6]. Although a chemical control of the disease is effective, the extensive use of chemicals leads to soil water pollution, is a threat to food safety and thus may directly harm human health but may also result in drug resistance in pathogens [7]. Biological control has less impact on the environment, does not promote drug resistance, and is relatively safe for humans and animals, which aligns well with sustainable agricultural development and the prospects for a broad application of biocontrol agents for the prevention of lily Fusarium wilt.

*Bacillus* species are bacteria with biocontrol activities that are capable of preventing plant diseases and play an important role in biocontrol, drug development, and food fermentation. The endospores formed by *Bacillus* spp. Are highly resistant to heat and desiccation; these properties enable their development into commercial products that can be easily stored and have a long shelf life [8,9]. *Bacillus* spp. Produce a range of antimicrobial substances that favor the colonization of crops, act as a biocontrol agent, and thus are ideal microorganisms for controlling crop diseases [10,11]. *Bacillus amyloliquefaciens* strain TB6 promotes root growth, increases the fresh weight of ginseng roots, and improves the activities of polyphenol oxidase and catalase [12]. *Bacillus pumilus* strain TUAT-1 promotes forage rice growth and nutrient uptake potential, and improves the soil properties (e.g., pH, total nitrogen, exchangeable ammonium-nitrogen (NH_4_^+^-N), and potassium contents) [13].

The rhizosphere soil is the site of materials and energy exchange between the plant body and the soil. The plant body influences the rhizosphere soil properties through respiration and the secretion of organic matter, and in turn, the soil is a source of nutrients for plants through the rhizosphere zone [14]. Soil microorganisms play important and diverse roles in the soil ecosystem and are essential functional components of the soil. They participate in the matter and energy cycles of the soil ecosystem, greatly impact plant growth, and are an indicator of soil fertility [15]. Exogenous bacteria introduced into the soil have been shown to alter the community structure of soil rhizosphere microorganisms by colonizing plant roots to a specific population density or remain in plant roots for a specific period [16,17]. When applied correctly, biocontrol bacteria improve soil quality and fertility to some extent, but their improper introduction may result in a soil microbiological imbalance.

In recent years, the application of *Bacillus* spp. for the control of plant pathogenic fungi has been studied intensively and many positive results have been reported. However, to date, research on the control of lily Fusarium wilt has been relatively limited, and the mechanism of the antagonistic activity of *Bacillus subtilis* towards *Fusarium* spp. has not been fully elucidated.

This study aimed to comprehensively investigate the effects of applying *B. amyloliquefaciens* BF1 and *B. subtilis* Y37 on the physicochemical properties and structural diversity of microbes in the rhizosphere soil. We determined the physicochemical properties of the rhizosphere soil under four treatments and sequenced the microbial community in the rhizosphere soil using high-throughput sequencing. The FunGuild and Tax4Fun tools were used for a functional prediction of soil fungi and bacteria. The ultimate goal was to improve the utilization of *Bacillus* strains as biocontrol agents to prevent plant diseases in crops.

## 2. Materials and Methods

### 2.1. Strains

Cultures of the bacteria *B. amyloliquefaciens* strain BF1 and *B. subtilis* strain Y37, which can inhibit the proliferation of many pathogenic fungi and control lily Fusarium wilt disease, were preserved in the laboratory of the Institute of Bast Fiber Crops, Chinese Academy of Agricultural Sciences (Changsha, China). A single colony each of BF1 and Y37 activated on nutrient agar overnight was inoculated in a lysogeny broth (LB) and incubated overnight at 30 °C while shaking at 180 rpm. Fresh LB medium was then inoculated with a 10% volume of the bacterial suspension and cultured at 30 °C while shaking at 180 rpm for 48 h. The BF1 and Y37 broth cultures were then diluted with sterile water to 1.0 × 10^8^ colony forming units (CFU)/mL for the irrigation of lily bulbs.

### 2.2. Experimental Design

Lily bulbs were planted at the Experimental Station Farm in Yuanjiang City, Hunan Province, China, where they had been planted for three consecutive years with an incidence of Fusarium wilt of more than 50% per year. Four treatments were applied in the experiment: BG (no lily bulbs planted), CK (lily bulbs planted, no bacterial treatment), BF1 (lily bulbs planted, irrigated with BF1 suspension), and Y37 (lily bulbs planted, irrigated with Y37 suspension). The four different treatments contained four replicates each with a treatment area of 27 m^2^ and each replicate was separated by a 3 m wide protection row. After planting, each lily bulb in the BF1 and Y37 treatments was irrigated with 40 mL of the BF1 and Y37 suspensions, respectively. A second irrigation treatment was applied 1 week after planting. Field management was consistent with local practices. After 4 weeks of growth, based on the percentage of the bulb area affected by Fusarium wilt, the disease severity was assigned to one of six classes: class 0 = 0%, class 1 = 1–20%, class 3 = 21–40%, class 5 = 41–60%, class 7 = 61–80%, and class 9 = 81–100%. The disease index (DI) for lily Fusarium wilt incidence was calculated with Equation (1),
(1)$$\mathrm{DI}\;(\%)=\sum_{i=0}^{9}\left(n_{i}\;\times \;i\right)\;\times \;\sum_{i=0}^{9}{{(n}_{i})}^{-1}\times \;100$$
where *i* is the severity class and n*_i_* is the number of plants assigned to class *i*. The DI values calculated ranged from 0 (no disease) to 100 (highest disease level). The disease reduction (DR; %) was calculated with the formula:DR (%) = (DI-ck − DI-test)/DI-ck × 100(2)

In addition, plant height, stem thickness, fresh weights (excluding seed bulbs), and bulbs’ yield were determined 70 days after the second irrigation treatment.

### 2.3. Rhizosphere Soil Sample Collection

In each replicate, the five-point sampling method was used to take 6 lilies at each point, and these 30 lilies’ diseases were investigated (4 weeks after the second irrigation treatment). The same method was used to sample and measure growth indicators such as the height, stem thickness, and fresh weight of the 30 lilies. The same five-point sampling method was used for each replicate, with two lilies taken at each point, and the inter-root soil around the roots of 10 lilies was carefully collected with a brush after shaking off large clumps of soil (70 days after the second irrigation treatment). Collected samples were placed in sterile polyethylene bags and immediately transported to the laboratory in ice boxes (70 days after the second irrigation treatment). After homogenization, the soil samples were passed through a 2 mm sieve to remove plant residues and stones and were then divided into two subsamples. One subsample (2 g per replicate) was stored at −80 °C for microbial DNA analysis; the other subsample (20 g per replicate) was air-dried for a physicochemical analysis.

### 2.4. Determination of Rhizosphere Soil Physicochemical Properties

To investigate the influence of the two *Bacillus* spp. On soil physicochemical properties, the pH, total nitrogen (TN), total potassium (TK), total phosphorus (TP), available potassium (AK), available phosphorus (AP), organic phosphorus (OP), organic carbon (OC), ammonium-nitrogen (NH_4_^+^-N), and nitrate-nitrogen (NO_3_^−^-N) contents were determined for each sample. The pH was determined using the potentiometric method. The TN content was measured using ultraviolet spectrophotometry following alkaline potassium persulfate digestion [18]. The TK content was determined using the sodium hydroxide melting method and the OC content was quantified using the potassium dichromate oxidation–spectrophotometric method [19,20]. The concentration of phosphate ions in the soil was determined using the molybdenum blue method, and the TP and OP contents were determined using the Mo-Sb antispectrophotometric method; the NH_4_^+^-N content was quantified using the indophenol-blue colorimetric method; the NO_3_^−^-N content was determined using the nitro-salicylic acid method; the AK content was quantified using tetraphenylboron-sodium turbidimetry method; and the AP content was measured by the molybdenum antimony anticolorimetric method. The TP, NH_4_^+^-N, NO_3_^−^-N, AK, AP, and OP contents were measured using corresponding kits obtained from ZCIBIO Technology Co., Ltd. (Shanghai, China) following the manufacturer’s instructions.

### 2.5. DNA Extraction, PCR Amplification, and High-Throughput Sequencing

Total genomic DNA was extracted using the hexadecyltrimethylammonium bromide method [21]. The purity and concentration of DNA was checked by agarose gel electrophoresis. Using the diluted genomic DNA as the template, PCR was performed using gene-specific primers and a sample-specific barcode with the Phusion^®^ High-Fidelity PCR Master Mix with GC Buffer (New England Biolabs, Ipswich, MA, USA) and high-efficiency, high-fidelity enzymes in accordance with the selected sequencing regions to ensure the amplification efficiency and accuracy. The diluted genomic DNA was used as the template to amplify the fungal ITS1 region with the primers ITS5-1737F (5′-GGAAGTAAAAGTCGTAACAAGG-3′) and ITS2-2043R (5′-GCTGCGTTCTTCATCGATGC-3′). The V3–V4 variable region of bacterial 16S rRNA was amplified with the primers 314F (5′-CCTAYGGGRBGCASCAG-3′) and 806R (5′-GGACTACNNGGGTATCTAAT-3′). The PCR products were detected by electrophoresis using 2% agarose gels; the PCR products were mixed in equal amounts according to their concentrations and then purified by electrophoresis using 2% agarose gels in 1× TAE buffer. The target bands were recovered by shearing. The PCR product was then purified from the gel using the GeneJET Gel Extraction Kit (Thermo Fisher Scientific, Waltham, MA, USA). The cDNA libraries were constructed using the Ion Plus Fragment Library Kit 48 rxns library building kit (Thermo Fisher Scientific, Waltham, MA, USA). After the constructed library was checked by Qubit quantification and library testing, sequencing was conducted using the Ion S5™ XL System (Thermo Fisher Scientific, Waltham, MA, USA). High-throughput sequencing was performed by Novogene Co., Ltd. (Beijing, China).

### 2.6. Data Analysis

Using Cutadapt (v1.9.1) [22], quality control was performed on the raw reads to remove the adapter sequences and low-quality reads, and then the barcode and primer sequences were removed. The retained raw reads required further processing to remove chimeric sequences; the read sequences were compared with the species annotation database to detect chimeric sequences, which were removed to obtain the final clean reads [23,24].

All clean reads of all samples were clustered using Uparse (v7.0.1) [25]. By default, the sequences were clustered into operational taxonomic units (OTUs) with 97% identity. Representative sequences for the OTUs were selected by the UPARSE-OTU algorithm, based on the sequences with the highest frequency of occurrence in each OTU. Species annotation was performed on OTUs sequences, and species annotation analysis was performed using the Mothur method [26] with the SSUrRNA database [27] of SILVA132 (with a threshold value of 0.8 to 1) to obtain the taxonomic level: kingdom, phylum, class, order, family, genus, and species. In addition, the data for each sample were homogenized, using the optimal minimum number of sequences per sample as the criterion for homogenization. Alpha- and beta-diversity analyses were performed on the homogenized data. The Richness, ACE, Chao1, and other alpha diversity indices of each sample were calculated using Qiime (v1.9.1).

The ecological functions of fungi were categorized and a FunGuild database was constructed. Based on the species information obtained from the amplicon analysis, the ecological functions of species in the environment already available in the literature was queried. Function prediction with the Tax4Fun tool was performed using the nearest-neighbor method based on the minimum *16S* rRNA sequence similarity. This was determined by extracting the whole-genome *16S* rRNA gene sequences of prokaryotes from the Kyoto Encyclopedia of Gene and Genomes (KEGG) database and comparing them to the SILVA SSU Ref NR database using the BLASTN algorithm (BLAST bitscore > 1500) to generate a correlation matrix. Function annotations from the KEGG database were used to provide prokaryotic whole-genome functional information and assigned to sequences using the UProC and PAUDA tools and compared with the SILVA database to achieve functional annotations consistent with the SILVA database. The sequenced samples were clustered into OTUs using the SILVA database sequences as reference sequences to obtain the functional annotation information.

A statistical analysis was conducted with R v4.2.1. An analysis of variance was performed to determine significant differences using the “aov” function. A simple clustering heatmap of the dominant soil microbe species was generated using the “heatmap” R package. A principal coordinate analysis (PCoA) was conducted to detect dissimilarity among the soil microbe communities based on the Bray–Curtis distance. A redundancy analysis (RDA) was performed to investigate the effect of soil physicochemical properties on soil microbe community structure using the “vegan” R package. Data were visualized using the “ggplot2” R package.

## 3. Results and Discussion

### 3.1. Biocontrol Activity of Bacillus BF1 and Y37 against Fusarium Wilt of Lily

Lily plants irrigated with the *Bacillus* suspensions had a significantly increased height, stem thickness, fresh weights, and yield compared with those of the CK group. The control efficacies of the Bacillus strains BF1 and Y37 against Fusarium wilt of lily were 58.74% and 68.93%, respectively (Table 1). *Bacillus subtilis* and *B. amyloliquefaciens* play an important role in preventing plant diseases by various mechanisms, such as competition, antagonism, the activation of chitinase and polyphenol oxidase, the induction of host-plant resistance, or the promotion of plant growth [28,29]. However, their colonization of the rhizosphere is a prerequisite for these bacteria to exert their specific functions [30]. In previous studies, *B. amyloliquefaciens* Y1 and *B. subtilis* IAGS174 exerted strongly inhibitory effects on Fusarium wilt of tomato and promoted plant growth through various mechanisms, such as biochemical, histological, and molecular re-regulation [29,31]. *Bacillus subtilis* IBFCBF-4, isolated and screened from the rhizosphere soil, showed a 66.95% inhibition of watermelon wilt [32]. In the present study, the irrigation of lily plants with the *Bacillus* strains BF1 and Y37 was effective in reducing the incidence of lily Fusarium wilt. In addition, the height, stem thickness, and yield of the lily plants increased significantly. These findings were consistent with previous studies in which *B. amyloliquefaciens* and *B. subtilis* were used to control blight and promote the growth of various crops, such as cucumber, pepper, and flax [4,33,34].

To further explore the mechanism by which Fusarium wilt pathogenicity was reduced, we analyzed the rhizosphere soil microbial community and properties of each group and explored its potential biocontrol mechanisms.

### 3.2. Effects of Bacillus BF1 and Y37 on Fungal Community Structure and Diversity in the Rhizosphere Soil

Using the Ion S5 XL sequencing platform, we obtained 1,251,367 valid ITS sequences with an average length of 247 bp from the 16 soil samples. Compared with the BG group, planting with lily bulbs increased the richness, ACE, and Chao1 indices of the rhizosphere soil, but no significant difference was observed for the BF1 and Y37 treatments compared with those of the CK group, suggesting that Bacillus had no significant effect on the diversity of fungi in the rhizosphere soil (Table 2). Similarly, a metagenomic sequencing and subsequent comparison of the deduced taxonomic profiles of the combined samples indicated that the application of *B. amyloliquefaciens* FZB42 did not have pronounced effects on the composition of the microbial communities in the rhizosphere soil of lettuce [35].

The OTUs were organized into 12 fungal taxonomic groups at the phylum level, accounting for 30.25–70.66% of the total fungi across all samples. *Ascomycota* constituted the greatest proportion (17.82–58.43%), followed by *Basidiomycota* and *Mortierellomycota* in all samples (Figure 1 and Appendix A). Liu et al. [9] similarly reported that *Ascomycota* was the major phylum, accounting for 68.75–78.30% of the total fungal OTUs, obtained from wheat rhizospheric soil samples, followed by *Mortierellomycota* and *Basidiomycota*, which is consistent with the present results. These findings imply a similarity in the structure of the rhizospheric fungal communities of plants growing in the field or a greenhouse, and also among different plant species and soil types, suggesting that the rhizospheric fungal community structure shows a high innate stability [36].

At the genus level, the dominant *Fusarium* was detected in all soil samples, but its relative abundance was decreased both after planting lily bulbs and after irrigation with *Bacillus* strain Y37. Conversely, the relative abundance of *Fusarium* was slightly increased by irrigation with the strain BF1. The BF1 and Y37 treatments increased the relative abundance of *Colletotrichum*, *Humicola*, *Solicoccozyma*, *Penicillium*, *Myceliophthora*, and *Chaetomium*. In addition, Y37 significantly increased the relative abundance of *Talaromyces*, but this effect was not detected after irrigation with BF1 (Appendix A and Figure 1). Particular *Colletotrichum* endophytes confer protective benefits to cacao hosts by reducing the disease incidence and damage caused by other plant pathogens [37,38]. Members of the *Humicola* genus are a rich source of unique and structurally diverse metabolites that show various bioactivities [39]. *Penicillium* species are mostly saprophytic in nature and numerous species are of particular value to humans. Probably the best known is *Penicillium notatum*, the source of the antibiotic penicillin [40]. Some *Talaromyces* species are well-known for the preparation of chiral building blocks or biotransformations and are utilized in pest biocontrol [41]. Therefore, we concluded that the irrigation of lily with BF1 and Y37 effectively increased the relative abundance of beneficial fungal genera and decreased the abundance of pathogenic fungal genera in soil, thereby augmenting their biocontrol effect.

A principal coordinate analysis is a nonbinding data-dimension-reduction analytic method that can be used to study the similarities or dissimilarities in community composition among samples. The PCoA axis 1 explained 23.55% and the PCoA axis 2 accounted for 14.18% of the total variance (Figure 2). With a PCoA, the closer the distance between samples, the more similar the community composition. The present results showed that the composition of all microbial communities in the rhizosphere soil samples was similar. These results demonstrated that irrigation with either BF1 or Y37 did not influence the fungal diversity in the lily rhizosphere soil. The reason may be that the effects of the *Bacillus* strains were attenuated by the high microbial diversity in the rhizosphere soil, which would be consistent with the findings of Han et al. [42] and Araujo et al. [36].

At the genus level, a total of 331 fungal genera were detected. Based on the OTUs and a taxonomic heatmap analysis, a cluster analysis was performed and a heatmap was generated for the 30 most common genera detected (Figure 2). The BG group samples clustered together, whereas the samples of the CK, BF1, and Y37 groups were not strongly clustered, indicating that the application of the *Bacillus* strains BF1 and Y37 did not significantly affect the fungal community structure and diversity in the rhizosphere soil. This finding was consistent with the PCoA results.

### 3.3. Effects of Bacillus BF1 and Y37 on Bacterial Community Structure and Diversity in the Rhizosphere Soil

Using the Ion S5 XL platform for sequencing, 1,203,663 effective bacterial sequences with an average length of 408 bp were obtained from the 16 soil samples. The richness, Shannon, Simpson, ACE, and Chao1 indices in the CK group were significantly higher than those of the BG group. This result indicated that planting lily increased the bacterial community richness, diversity, and evenness in the rhizosphere soil. In addition, the richness, Shannon, ACE, and Chao1 indices in the BF1 and Y37 groups were increased compared with those of the CK group, but these indices increased significantly after irrigation with BF1 but not with Y37. Thus, irrigation with the Bacillus strains increased the soil bacterial community richness and evenness (Table 3). These results are consistent with those of You et al. [43], who reported that treatment with *Bacillus subtilis* Tpb55 increased soil bacterial diversity relative to that of the control and fungicide treatment. In addition, Zhao et al. also found that the diversity of soil bacteria and fungi in the rhizosphere of cotton increased under *Bacillus subtilis* NCD-2 (BS) treatment [44]. Aside from being applied alone to alter the abundance and diversity of rhizosphere soil microorganisms, however, *Bacillus subtilis* can also be used as a carrier with biochar to prepare microbial biochar formulations. For example, it was found that BCMs treatment exhibited a significant increase in the abundance of bacterial genera in the rhizosphere soil of radish [45]. However, this study is contrary to the result that chemical fumigation reduces soil microbial diversity, which suggests that the application of biocontrol agents protects soil microbial diversity better than chemical fumigation [46,47].

The OTUs were organized into 39 bacterial taxonomic groups at the phylum level, which accounted for 93.60–98.14% of the total bacteria detected across all samples. The phyla *Proteobacteria*, *Acidobacteria*, *Chloroflexi*, *Actinobacteria*, and *Bacteroidetes* were the five most abundant phyla, all of which were detected in each soil sample (Figure 3 and Appendix A). Similarly, *Proteobacteria*, *Actinobacteria*, *Chloroflexi*, and *Acidobacteria* were the dominant phyla in a tea plantation area of Sichuan Agricultural University in China, of which *Proteobacteria* was the most dominant phylum [48]. The planting of lily increased the relative abundance of *Actinobacteria*. Irrigation with the *Bacillus* strains BF1 and Y37 decreased the relative abundance of *Verrucomicrobia*, and BF1 increased the relative abundance of *Actinobacteria* and *Firmicutes*.

At the genus level, a comparison of the relative abundances of the 20 most abundant genera revealed that four genera showed significant differences among the treatments (Figure 4). The planting of lily significantly decreased the relative abundance of *Bryobacter* and *Chujaibacter* and increased that of *Candidatus_Solibacter* and *Pseudolabrys*. Irrigation with *Bacillus* strains BF1 and Y37 increased the relative abundance of *Unidentified_Burkholderiaceae* and *Bradyrhizobium* and decreased that of *Pseudolabrys*. Thus, planting lily and irrigation with BF1 and Y37 stimulated the proliferation of certain bacteria but inhibited the abundance of other bacteria. Planting lily did not significantly affect the relative abundance of *Bacillus* in the soil, whereas the application of BF1 and Y37 significantly enhanced *Bacillus* abundance, indicating that irrigation with BF1 and Y37 promoted the abundance of *Bacillus* in the rhizosphere soil. This finding suggests that BF1 and Y37 effectively enrich beneficial bacteria in the lily rhizosphere (Appendix A and Table 4). Previous studies have shown that Bacillus affects the dominant soil flora through colonization or change in the community structure of soil microorganisms by secreting bacteriostatic substances [16,17].

In the PCoA of the bacterial community data, the first and second principal coordinates together explained 50.06% of the total variance, of which PCoA axis 1 accounted for 35.14% of the variance (Figure 4). Planting lily did not significantly affect the bacterial community compared with that of the BG group. However, irrigation with both *Bacillus* strains resulted in a separation of the BF1 and Y37 groups from the CK group primarily on axis 1.

At the genus level, a total of 524 genera were detected. Based on the OTUs and a taxonomic heatmap analysis, a cluster analysis was performed and a heatmap was generated for the 30 most common genera detected (Figure 4). The samples for each treatment group were strongly clustered, indicating that planting lily and application of the *Bacillus* strains BF1 and Y37 strongly affected the bacterial community structure in the lily rhizosphere soil, which differed from the present findings on fungal community structure. The reason for this may be that bacteria in the soil are more active than fungi and are susceptible to the application of foreign biocides. The PCoA and heatmap results both demonstrated that irrigation with BF1 or Y37 influenced the bacterial diversity of the lily rhizosphere soil, which is consistent with the findings of Han et al. [42] and Wan et al. [49].

### 3.4. Effects of Bacillus BF1 and Y37 on Soil Physicochemical Properties

Compared with the BG treatment, the CK treatment showed significantly decreased TP, NH_4_^+^-N, NO_3_^−^-N, AK, and AP contents. In particular, the AP content was decreased by 40.43% (Table 5). In addition, the TP and NH_4_^+^-N contents were dramatically decreased by 32.55% and 32.08%, respectively, which may be because the lily plants especially utilized these factors to enhance their growth. Conversely, the BF1 and Y37 treatment groups both showed effectively increased TP (increases of 107.59% and 98.63%, respectively), NH_4_^+^-N (increases of 128.70% and 141.44%, respectively), NO_3_^−^-N (increases of 56.46% and 31.03%, respectively), AK (increases of 40.21% and 34.55%, respectively), AP (increases of 145.76% and 155.80%, respectively), and OP (increases of 100.11% and 98.33%, respectively) contents compared with those of the CK. Plant growth-promoting rhizobacteria can inhibit pathogens by accelerating the decomposition of organic substances and dissolving the phosphorus and potassium in the soil to promote nutrient absorption by plants [50,51]. In the present study, the pH in all treatments was within the range of 7.0–7.1, indicating that the soil acid–base status remained neutral.

A redundancy analysis was performed to examine the relationship between the soil edaphic properties and the diversity of microbial communities in the four groups of soil samples. The effect of the 10 edaphic variables on the diversity of fungal communities in the soil in the four treatments, ranked from largest to smallest, was TK, pH, OC, NH_4_^+^-N, TN, AP, OP, TP, NO_3_^−^-N, AK, OC (Figure 5), and that for the bacterial communities was NO_3_^−^-N, TP, NH_4_^+^-N, pH, AP, OP, AK, TN, TK, OC (Figure 5). Fusarium wilt was positively correlated with pH, OC, and AP contents, and negatively correlated with the other edaphic variables. *Bacillus* was positively correlated with most of the physicochemical properties of the rhizosphere soil but was negatively correlated with TK and TN contents. The application of BF1 and Y37 significantly increased the contents of most edaphic variables compared with those of the CK group. Fusarium wilt was negatively correlated with most of the edaphic properties, whereas *Bacillus* was positively correlated with most of the physicochemical properties, indicating that the application of the two *Bacillus* strains did not favor *Fusarium* growth and thus the disease incidence decreased. These findings are consistent with a previous study that integrated biological control methods and effectively suppressed tobacco bacterial wilt by regulating the soil physicochemical properties, and promoting beneficial and antagonistic bacteria in the rhizosphere soil [52]. The longer arrows in the RDA ordinations (Figure 1) indicated that pH had a stronger effect on bacteria and fungi than the other physicochemical properties, which is consistent with the findings of Shi et al. [53] and Yu et al. [54].

### 3.5. Predictive Functional Profiling of the Rhizosphere Soil Microbial Community

Previous studies have shown that exposure to *B. subtilis* bioorganic fertilizer significantly enhanced the metabolism and absorption pathways for mineral elements and the sulfur/phosphonate/phosphinate metabolism [55]. We sought to verify whether BF1 and Y37 had the same effect. To construct functional profiles of the microflora, the rhizosphere soil fungal and bacterial ecological functions of the treatment groups were predicted with the FunGuild and Tax4Fun tools, respectively. Ten trophic modes of fungi were identified among the treatments. These comprised pathogen-saprotroph-symbiotroph (0.001–0.006%), pathotroph (1.365–2.459%), pathotroph-saprotroph (2.053–3.391%), pathotroph-saprotroph-symbiotroph (0.983–1.836%), pathotroph-symbiotroph (2.451–12.372%), saprotroph (6.232–18.502%), saprotroph-pathotroph-symbiotroph (0.003–0.048%), saprotroph-symbiotroph (0.000–0.005%), symbiotroph (0.291–2.999%), and unassigned (66.777–77.931%) (Figure 6). Except for the unassigned category, saprotroph was the dominant trophic mode and the frequency for the Y37 treatment was larger than that for the other treatments. In addition, the pathotroph mode was significantly less frequent in the BF1 and Y37 treatments compared with that of the CK. The dominant groups of functional soil fungi were endophyte plant pathogen, undefined saprotroph, undefined saprotroph-wood saprotroph, and plant pathogen-soil saprotroph-wood saprotroph, and the average abundance of all treatments accounted for 7.90%, 6.65%, 3.39%, and 2.54%, respectively (Figure 6). The abundance of dung saprotroph-undefined saprotroph-wood saprotroph in the BF1 treatment was increased by 102.64% compared with that of the CK. The abundance of undefined saprotroph-wood saprotroph in the Y37 treatment was increased by 315.58% over that of the CK. In addition, the BF1 and Y37 treatments showed a significantly increased abundance of undefined saprotroph compared with that of the CK.

Compared with the BG group, amino acid metabolism, glycerolipid metabolism, and fatty acid biosynthesis microbial activities were significantly upregulated in the CK treatment, whereas other metabolic pathways, such as sulfur metabolism, prokaryotic defense system, and five other microbial activities were significantly downregulated (*p* ≤ 0.05). Compared with the CK treatment, microbial activity in messenger RNA biogenesis, glycolysis/gluconeogenesis, carbon fixation pathways in prokaryotes, and citrate cycle (TCA cycle) were significantly more abundant in the BF1 treatment (*p* < 0.05). Additional metabolic pathways, such as alanine, aspartate, and glutamate metabolism, butanoate metabolism, cysteine and methionine metabolism, and methane metabolism, were upregulated in the BF1 treatment (Figure 7). Glycolysis/gluconeogenesis, glycerolipid metabolism, and the adipocytokine signaling pathway were significantly more abundant in the Y37 treatment compared with the CK treatment.

These results showed that irrigation with *Bacillus* strains BF1 and Y37 significantly upregulated the metabolism and absorption pathways of glycolysis/gluconeogenesis, the citrate cycle (TCA cycle), glycerolipid metabolism, and additional pathways. Thus, the application of the two *Bacillus* strains stimulated the proliferation of other beneficial microorganisms. A previous study reported that the application of the biocontrol strain BL12 significantly increased the relative abundance of certain KEGG pathways [56], which was consistent with the predictions in the present study.

## 4. Conclusions

The application of *B. amyloliquefaciens* BF1 and *B. subtilis* Y37 effectively controlled lily Fusarium wilt disease with control efficacies of 58.74% and 68.93%, respectively. Both strains significantly increased the bacterial richness and Shannon diversity indices in the rhizosphere soil of lily. Irrigation with BF1 and Y37 significantly increased the abundance of *Bacillus* in the soil, confirming the ability of the BF1 and Y37 strains to colonize the lily rhizosphere, and enhanced the abundance of beneficial microbes. In addition, the BF1 and Y37 strains influenced the soil edaphic properties. Interestingly, the RDA results showed that the application of the two *Bacillus* strains did not favor the growth of *Fusarium*, thus further reducing the incidence of lily Fusarium wilt disease. Certain metabolic pathways, such as glycolysis/gluconeogenesis, were upregulated after the application of the two *Bacillus* strains. This research provides important insights for understanding the mechanism of effective disease prevention by the *Bacillus* strains BF1 and Y37.

While this experiment investigated the effects of two different *Bacillus* species on the growth of lilies and their rhizosphere soil microbial community structure, the experiment was conducted at only one site, the Experimental Station Farm in Yuanjiang City, Hunan Province, China. Further field experiments at different sites are therefore required to support the findings of this study and to verify the results obtained.

## Figures and Tables

**Figure 1 microorganisms-11-01229-f001:**
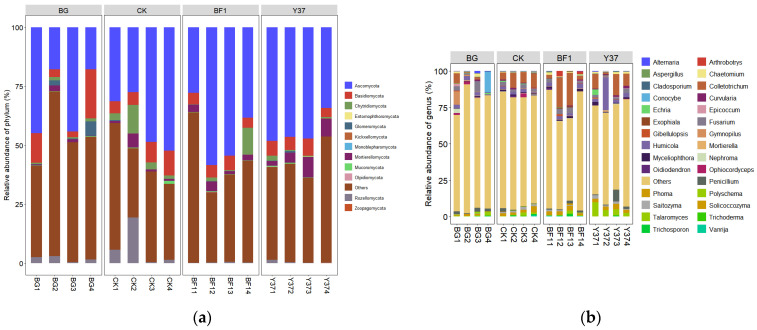
Relative abundance of fungal community structure in lily rhizosphere soil samples. (**a**) At the phylum level, (**b**) top 30 at the genus level. BG, no lily planted; CK, lily planted, no bacterial treatment; BF1, lily planted, irrigated with a 40 mL BF1 suspension; Y37, lily planted, irrigated with a 40 mL Y37 suspension.

**Figure 2 microorganisms-11-01229-f002:**
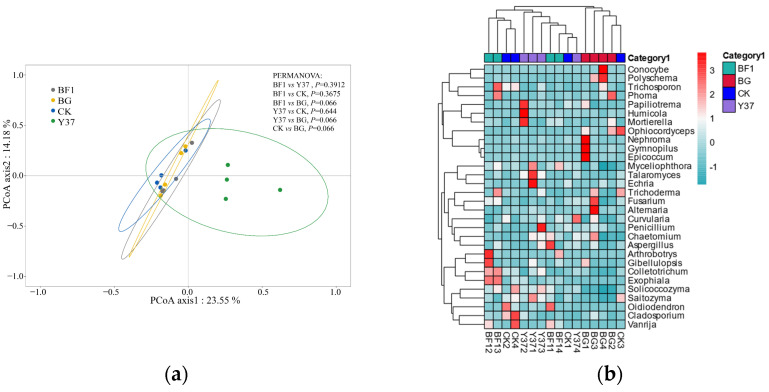
Principal coordinate analysis and hierarchical cluster analysis of the fungal community from lily (*Lilium brownii* var. *viridulum*) rhizosphere soil samples. (**a**) Principal coordinate analysis ordination of axis 1 and axis 2. (**b**) Heatmap of fungal communities based on the levels of the 30 most abundant genera in each sample. Blue and red shading indicate low and high relative abundances of fungal genera in the group, respectively. BG, no lily planted; CK, lily planted, no bacterial treatment; BF1, lily plant, irrigated with a 40 mL BF1 suspension; Y37, lily planted, irrigated with a 40 mL Y37 suspension.

**Figure 3 microorganisms-11-01229-f003:**
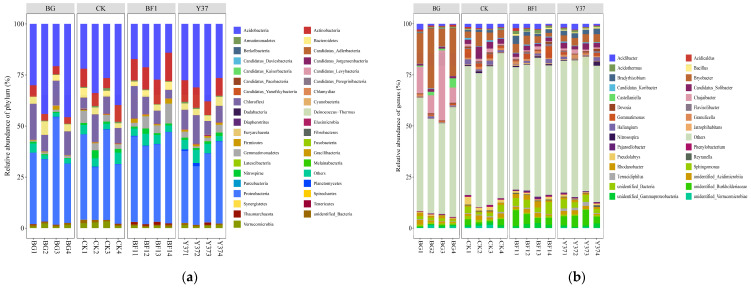
Relative abundance of bacterial community structure in lily rhizosphere soil samples. (**a**) Top 30 at the phylum level, (**b**) top 30 at the genus level. BG, no lily planted; CK, lily planted, no bacterial treatment; BF1, lily planted, irrigated with a 40 mL BF1 suspension; Y37, lily planted, irrigated with a 40 mL Y37 suspension.

**Figure 4 microorganisms-11-01229-f004:**
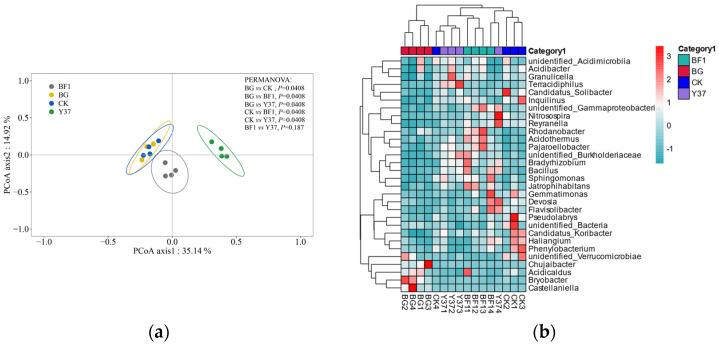
Principal coordinate analysis and hierarchical cluster analysis of the bacterial community from lily (*Lilium brownii* var. *viridulum*) rhizosphere soil samples. (**a**) Principal coordinate analysis ordination of axis 1 and axis 2. (**b**) Heatmap of the bacterial communities based on the levels of the 30 most abundant genera in each sample. Blue and red represent low and high relative abundances of bacterial genera in the group, respectively. BG, no lily planted; CK, lily planted, no bacterial treatment; BF1, lily planted, irrigated with a 40 mL BF1 suspension; Y37, lily planted, irrigated with a 40 mL Y37 suspension.

**Figure 5 microorganisms-11-01229-f005:**
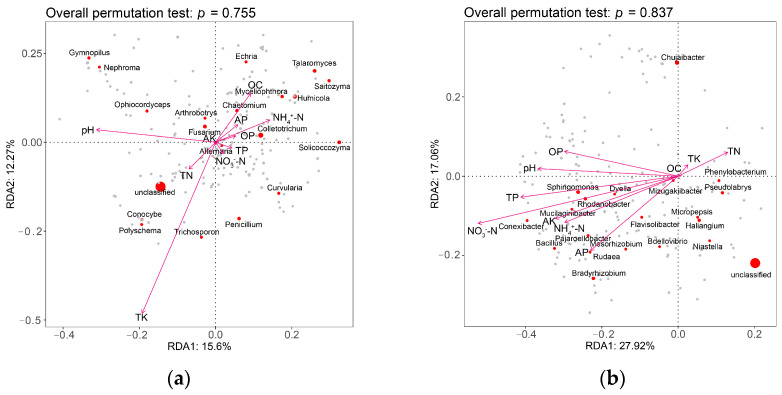
Redundancy analysis (RDA) ordinations showing the effects of soil physicochemical properties on microbial communities in the rhizosphere soil of lily (*Lilium brownii* var. *viridulum*). (**a**) Fungi, (**b**) bacteria.

**Figure 6 microorganisms-11-01229-f006:**
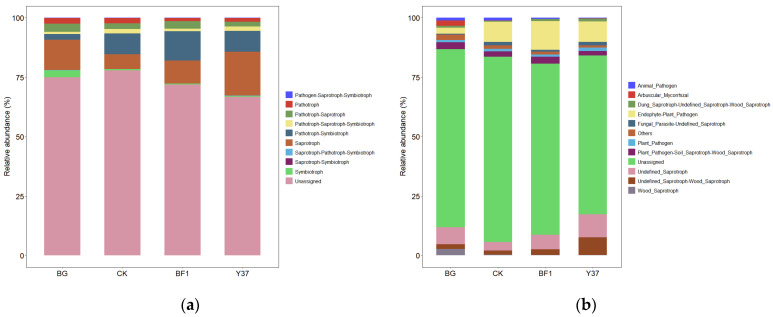
Relative abundance of fungal trophic groups (**a**) and functional groups (**b**). BG, no lily planted; CK, lily planted, no bacterial treatment; BF1, lily planted, irrigated with a 40 mL BF1 suspension; Y37, lily planted, irrigated with a 40 mL Y37 suspension.

**Figure 7 microorganisms-11-01229-f007:**
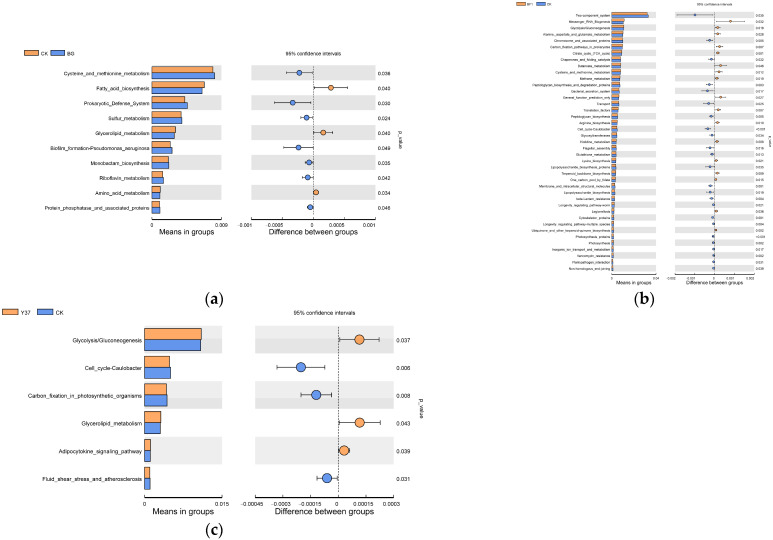
The functional profiles of bacterial communities (KEGG level 3). (**a**) CK vs. BG. (**b**) CK vs. BF1. (**c**) CK vs. Y37. BG, no lily planted; CK, lily planted, no bacterial treatment; BF1, lily planted, irrigated with a 40 mL BF1 suspension; Y37, lily planted, irrigated with a 40 mL Y37 suspension.

**Table 1 microorganisms-11-01229-t001:** Control efficacy of *Bacillus* strains BF1 and Y37 on Fusarium wilt of lily (*Lilium brownii* var. *viridulum*).

Treatment	Growth Indices			Yield (kg)	DI (%)	DR (%)
	Height (cm)	Stem Thickness (cm)	Fresh Weight (g)			
BG	–	–	–	–	–	–
CK	73.80 ± 6.38c	8.07 ± 0.98b	24.00 ± 8.61b	8.67 ± 3.25b	76.30 ± 4.63a	–
BF1	85.40 ± 1.14b	9.76 ± 0.43a	37.61 ± 10.83b	11.21 ± 3.31b	31.48 ± 3.90b	58.74 ± 5.11
Y37	96.20 ± 7.85a	10.66 ± 0.73a	60.07 ± 14.85a	17.73 ± 5.09a	23.70 ± 3.39b	68.93 ± 4.45

Note: BG, no lily planted; CK, lily planted, no bacterial treatment; BF1, lily planted, irrigated with a 40 mL BF1 suspension; Y37, lily planted, irrigated with a 40 mL Y37 suspension. DI, disease index; DR, disease reduction. Different lowercase letters within the same column indicate a significant difference between treatments (*p* < 0.05).

**Table 2 microorganisms-11-01229-t002:** Fungal community richness and diversity indexes of lily rhizosphere soil samples.

Treatment	Richness	Shannon	Simpson	ACE	Chao1	Coverage (%)
BG	638.50 ± 126.33b	6.45 ± 0.48a	0.966 ± 0.01a	762.70 ± 124.66b	780.30 ± 117.39b	99.62 ± 0.00a
CK	835.25 ± 90.15a	6.87 ± 0.22a	0.969 ± 0.01a	1002.44 ± 96.53a	1042.46 ± 99.65a	99.48 ± 0.00b
BF1	843.25 ± 84.83a	6.33 ± 0.69a	0.936 ± 0.05a	1020.11 ± 70.96a	1027.33 ± 78.74a	99.46 ± 0.00b
Y37	837.50 ± 40.17a	6.62 ± 0.44a	0.959 ± 0.02a	1004.25 ± 73.17a	1012.29 ± 86.99a	99.47 ± 0.00b

Note: BG, no lily planted; CK, lily planted, no bacterial treatment; BF1, lily planted, irrigated with 40 mL BF1 suspension; Y37, lily planted, irrigated with 40 mL Y37 suspension. Different lowercase letters within the same column indicate a significant difference between treatments (*p* < 0.05).

**Table 3 microorganisms-11-01229-t003:** Bacterial community richness and diversity indexes of lily rhizosphere soil samples.

Treatment	Richness	Shannon	Simpson	ACE	Chao1	Coverage (%)
BG	1186 ± 97c	6.79 ± 0.08c	0.943 ± 0.02b	1521 ± 90c	1531 ± 102c	98.82 ± 0.08a
CK	1469 ± 112b	8.35 ± 0.26b	0.991 ± 0.00a	1829 ± 119b	1851 ± 126b	98.66 ± 0.07ab
BF1	1755 ± 142a	8.74 ± 0.17a	0.994 ± 0.00a	2165 ± 216a	2170 ± 235a	98.42 ± 0.22c
Y37	1575 ± 182ab	8.44 ± 0.33ab	0.992 ± 0.00a	1971 ± 207ab	1982 ± 230ab	98.54 ± 0.14bc

Note: BG, no lily planted; CK, lily planted, no bacterial treatment; BF1, lily planted, irrigated with a 40 mL BF1 suspension; Y37, lily planted, irrigated with a 40 mL Y37 suspension. Different lowercase letters within the same column indicate a significant difference between treatments (*p* < 0.05).

**Table 4 microorganisms-11-01229-t004:** The *Bacillus* genus relative abundance of bacterial community structure in lily rhizosphere soil samples.

Treatment	*Bacillus* Genus Relative Abundance (%)
BG	0.057 ± 0.028b
CK	0.091 ± 0.044b
BF1	0.456 ± 0.295a
Y37	0.438 ± 0.103a

Note: BG, no lily planted; CK, lily planted, no bacterial treatment; BF1, lily planted, irrigated with a 40 mL BF1 suspension; Y37, lily planted, irrigated with a 40 mL Y37 suspension. Different lowercase letters within the same column indicate a significant difference between treatments (*p* < 0.05).

**Table 5 microorganisms-11-01229-t005:** Ten physicochemical indicators of different treatments.

	BG	CK	BF1	Y37
pH	7.06 ± 0.041ab	7.02 ± 0.017bc	7.07 ± 0.040a	7.01 ± 0.012c
TN (g/kg)	1.05 ± 0.303a	1.15 ± 0.305a	0.78 ± 0.096a	0.78 ± 0.042a
TK (%)	1.70 ± 0.002a	1.72 ± 0.001a	1.56 ± 0.001a	1.51 ± 0.001a
TP (μg/g)	1338.356 ± 170.327b	902.740 ± 176.785c	1873.973 ± 174.197a	1793.151 ± 172.545a
NH_4_^+^-N (μg/g)	12.0090 ± 2.594b	8.1571 ± 0.489c	18.6556 ± 2.710a	19.6941 ± 1.751a
NO_3_^−^-N (μg/g)	201.373 ± 10.083c	174.759 ± 12.734d	273.429 ± 14.243a	228.978 ± 12.465b
AK (mg/kg)	82.620 ± 1.118b	67.686 ± 11.880c	94.904 ± 2.944a	91.071 ± 3.585ab
AP (μmol/g)	24.767 ± 1.682b	14.754 ± 1.561c	36.259 ± 3.706a	37.740 ± 1.818a
OP (μg/g)	618.912 ± 119.298ab	405.518 ± 265.598b	811.473 ± 184.833a	804.262 ± 175.493a
OC (g/kg)	23.65 ± 2.073a	24.02 ± 4.116a	25.09 ± 5.774a	24.84 ± 3.964a

Note: BG, no lily planted; CK, lily planted, no bacterial treatment; BF1, lily planted, irrigated with a 40 mL BF1 suspension; Y37, lily planted, irrigated with a 40 mL Y37 suspension. TN, total nitrogen; TK, total potassium; TP, total phosphorus; AK, available potassium; AP, available phosphorus; OP, organic phosphorus; OC, organic carbon; NH_4_^+^-N, ammonium-nitrogen; and NO_3_^−^-N, nitrate-nitrogen. Different lowercase letters within the same row indicate a significant difference between treatments (*p* < 0.05).

## Data Availability

The data of this study are available from the correspondence author upon reasonable request.

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
