# Peer review of "Two Bacillus spp. Strains Improve the Structure and Diversity of the Rhizosphere Soil Microbial Community of Lilium brownii var. viridulum"

_microorganisms, 2023, doi:10.3390/microorganisms11051229_

Round 1

Reviewer 1 Report

This paper presents interesting data on the effects of two different Bacillus biocontrol strains on Fusarium wilt of lily, soil physicochemical properties, and soil microbial communities. The biggest issue and limitation of the study is that it consists of a single unrepeated experiment with limited replication. Although extensive analyses of the microbial communities are conducted, it is not clear how representative these changes may be to these treatments if the study were to be conducted again or under different conditions. Thus, additional documentation is needed to verify the results observed. Essentially this serves as a preliminary experiment, but results from this study can still be useful. However, at the very least, the authors must acknowledge the limitations and preliminary aspect of these results and conclusions and discuss these issues within the paper. In addition, there are numerous occurrences throughout the paper where the description provided is incomplete or inaccurate, and some re-organization is also needed. More detail is needed on some aspects of the methodology, particularly regarding sampling, replication, and experimental design. All results are based on only 4 samples from each treatment (and experiment is not repeated, or conducted in more than 1 location), which seems insufficient to make any firm conclusions regarding the soil microbial effects. I have attached an edited file containing additional comments and suggestions for revision, but the entire manuscript needs to be thoroughly revised before it can be accepted for publication. 

Reviewer 2 Report

The article is exceptionally well written, and the exhibit depicts the relative abundance of the bacterial and fungal communities in the soil surrounding the lily rhizosphere. In addition, additional bacterial functional profiles were expressed in the results section.

Both major leaf spot diseases and bacterial leaf scorch have a negative impact on lily plant growth and yield. Chemical fumigation can potentially reduce the total number of microorganisms and their diversity. Fungal sporulation and bacterial population were both reduced. The authors should discuss this point in the discussion.

Round 2

Reviewer 1 Report

Authors have adequately addressed previous reviewer comments and suggestions in their revised version of the paper. Thank you for making the requested revisions. The paper can now be accepted for publication.